# Visitors' support of ocean protection in a low-use marine protected area: Gray's reef national marine sanctuary

Marieke Lemmen[1], Robert C. Burns[1]*, Ross Andrew[1], Jasmine Cardozo Moreira[2], Danielle Schwarzmann[3], Mary Allen[4]

**1** School of Natural Resources and the Environment, West Virginia University, Morgantown, West Virginia, United States of America, **2** Universidade Estadual do Paraná, Ponta Grossa, Brazil, **3** National Oceanic and Atmospheric Administration, Office of National Marine Sanctuaries, Silver Spring, Maryland, United States of America, **4** Lynker, in support of NOAA Office of Coastal Management, Silver Spring, Maryland, United States of America

\* robert.burns@mail.wvu.edu

## Abstract

Marine protected areas (MPAs) are important tools for conserving biodiversity and supporting healthy ocean ecosystems and economies. However, the effectiveness of MPAs is dependent upon the support of those who use and are affected by them. Understanding how recreational users relate to protected areas can provide key insights into long-term support for conservation initiatives. This study focused on a low-use MPA in the Atlantic Ocean, 20 miles offshore from Savannah, Georgia, USA. The study used an online survey of recreational users, primarily anglers, to understand human nature interactions and perceptions of support for environmental protection in the MPA and surrounding coastal settings. Understanding these perceptions is important to the success of environmental protection and helps to parse out willingness to act to ensure a sustainable use of ocean resources. The results of a series of regression models highlight the complexity of human behavior, showing that support for protection and willingness to act were influenced by broader attitudes, such as environmental concern, in combination with coastal or non-coastal residency. These findings indicate the need for managers to consider and address both the broader public and the local communities affected by potential restrictions associated with a given area's protection status.

## 1. Introduction

Marine environments provide a variety of services to humans, such as different recreation opportunities, food supplies, energy supplies, research, and resources for a variety of use that people benefit from and depend on. Thus, the connections people have to the ocean, individually and as a society, are numerous. In

**Data availability statement:** All relevant data are within the paper and its Supporting Information files.

**Funding:** National Marine Sanctuary Foundation https://marinesanctuary.org/ Grant #19-04-B-210 Initials--RCB The funders did not play a role in data collection analysis, etc.

**Competing interests:** The authors have declared that no competing interests exist.

recent years, there has been a growth in ocean protection, particularly in so called "national waters," such as national marine sanctuaries [1]. Few if any marine areas are unaffected by human activities, which comes along with high potential for stressors to these environments [2]. As Heath [2] comments, the complexity of marine ecosystems cannot be understated, including complex human interactions. In fact, overuse and destruction of natural resources present a challenge to humanity and need attention [2–4]. However, as Lotze et al. [4] noted, "increased awareness and concern about ocean threats and protection could translate into changing of individual behaviors as well as regional, national and international stewardship and governance" (p. 21).

Marine environments and protected areas overall serve as touristic destinations [1,5], leading to a well-known paradox in tourism. People often choose these sites and destinations based on the desire to experience the natural qualities and attractiveness of the marine environment along with the recreational opportunities that are provided [6]. At the same time, the increased use of these areas results in negative impacts to the environment [2], which can negatively impact the visitor experience and the ecological characteristics and functions of the natural environment simultaneously. This illustrates the need for visitor use management and the protection of resources [7].

A common way to manage and protect resources and visitor impacts in marine environments is the implementation of marine protected areas (MPAs). Approximately 10% of coastal and marine areas worldwide are managed for conservation or protection [8]. The International Union for Conservation of Nature (IUCN) defines a protected area as a "geographical space, recognized, dedicated and managed, through legal or other effective means, to achieve the long-term conservation of nature with associated ecosystem services and cultural values" [9]. The restriction of specific activities and the limitation of visitor/user numbers reflect one common measure to achieve the goal of conservation. The United States includes over 600,000 square miles (155,400 square kilometers) of underwater protected areas designated as national marine sanctuaries. National marine sanctuaries are MPAs managed by the National Oceanic and Atmospheric Administration (NOAA) and characterized by their special conservation, recreational, ecological, historical, scientific, educational, cultural, archaeological, and aesthetic attributes and features [10]. The area examined in this study was Gray's Reef National Marine Sanctuary (GRNMS), a live-bottom reef approximately 20 miles (32 kilometers) off the coast of Georgia, as well as surrounding coastal and ocean areas.

How much use is acceptable in protected areas, how much negative impact is acceptable, when do we have to adjust the management and actions, and how can we protect the environment successfully? These are all essential questions related to managing and balancing use and protection of natural resources. Public support is crucial to successful management and conservation [11]. As people continue to create impacts on the environment, problems and solutions should include their input with respect to monitoring and implementation. To identify the best strategies for

impactful and beneficial management, understanding the visitors' perceptions and how they relate to support for protection, as well as actual environmentally friendly behaviors, is helpful [4,12,13].

The focus of this study was to identify the resource users of GRNMS and to achieve a better understanding of their environmental perceptions. Specifically, the research sought to understand the level support for protection in the study area. Humanity is facing challenges from overuse and destruction, and these require attention [3]. Stewardship represents a way people can help serve and protect the environment, while the environment provides services to people in the form of touristic places, resources to consume, etc. Therefore, knowing whether people support protection in the area is important. Furthermore, identifying the public's willingness to act can help define and implement evidence-based management decisions. The willingness of resource users to undertake action to ensure sustainable use of resources in the study area was assessed in a second step. This effort also sought to understand the drivers of supportive attitudes toward marine protection in the area. Overall, the primary purpose of this study was to examine the influences on the support of protection of ocean areas and resources of visitors to GRNMS, an offshore marine protected area.

As a basis for this study, the following research questions were posed:

$R_1$: Do users support the protection of the ocean and resources in coastal Georgia and in GRNMS?

$R_1a$: Do users support the protection of the ocean and resources in GRNMS?

$R_1b$: Do users support the protection of the ocean and resources in coastal Georgia?

$R_1c$: Are the users willing to act to ensure sustainable use of the resources?

$R_2$: What impacts drive resource-users' support for protection in and around GRNMS?

$R_3$: What impacts resource-users' willingness to undertake action to ensure sustainable use of ocean resources?

## 2. Background

### 2.1. Environmental values

Human nature interactions have been the subject of many studies in the past and present [14]. They are complex and can threaten environments due to negative effects of overuse and destruction [3], population growth, globalization processes [14], climate change, loss of biodiversity, and impacts to other ecosystem or environmental services, such as clean air, water, and processes of carbon sequestration [12,15]. Understanding the relationship between human actions and their underlying cognitive processes can lead to insights about human responses, including resistance to or acceptance of change, and the critical points at which innovation can arise [12].

### 2.2. Cognitive hierarchy

The cognitive hierarchy is a model that describes several levels of human cognition leading to action. These levels are a person's values, value orientations, ideologies, and attitudes, which lead to a behavioral intention, and finally to a specific behavior [12,16]. Researchers applied the model of cognitive hierarchy [17] in a survey study examining people's perceptions about human-wildlife interaction in western United States. In this model, attitudes are defined as the evaluative cognitions that are the immediate cause of human behavior, and values are the abstractions leading indirectly to behavior [12]. Generally, values can be understood as fundamental orientations, life goals, or guiding principles leading to beliefs, attitudes, and guiding behavior [12]. Manfredo et al. [12] use the term "value orientations" to reflect the influence of ideology in the cognitive hierarchy and distinguish between the mutualism wildlife value orientation and domination value orientation. Mutualism is based on an egalitarian ideology, where animals and people both have rights and they share equal moral status [12]. People with this ideology are more likely to engage in welfare-enhancing behaviors for individual wildlife and less likely to cause damage or death to an animal. On the other hand, the domination value orientation leads more likely to actions prioritizing human well-being over wildlife [12]. Changes in lifestyle and circumstances can cause a shift from one value orientation to another [12]. Moreover, these value orientations are applicable to other natural resources,

such as marine resources and MPAs. The human dimensions of environmental issues overall can be approached by using this theory and examining biocentric attitudes versus egoistic or anthropocentric attitudes [18]. Depending on the position in this continuum, people can vary from strong environmentalists with deep concern about conservation to those with apathy toward conservation issues and a view of nature as a source of resources to be used solely for the benefit of human development [19,20]. For this study, the concept of value orientation can be assessed through surveys. Participants can be asked to rate their environmental concerns, environmental values, and willingness to engage in sustainable behavior. Therefore, it is expected that respondents' ratings of values, their environmental concern levels, and their support of protection show similar trends and statistical relationships. The cognitive hierarchy model has been used in past research to help define expected behaviors and attitudes based upon defined influencing factors, such as socio-demographics, concern levels, and environmental values.

### 2.3. Environmental and marine protection perception

Understanding environmental values is important to the success of environmental protection [21]. A person's values and attitudes about environmental issues are often understood as the foundation of support for environmental protection, meaning that a specific value orientation could potentially lead to more environmentally friendly behavior than others [12,14,18,22]. In terms of public perception of marine protection, studies have shown that there are general similarities between perceptions of ocean threats and protection across different countries and regions [4]. In past studies, most people (over 70%) indicated that they believe human activities negatively impact the health of marine environments, and also were found to be supportive of ocean protection in their region [4]. Highest perceived threats to the ocean's health are pollution, fishing, habitat alteration, climate change, and/or loss of biodiversity, among others [4].

### 2.4. Ocean protection

The Secretariat of the UN Convention on Biological Diversity has released a first draft of a new global biodiversity framework to guide actions to preserve and protect nature and its essential services to people [23]. As of 2020, approximately 10% of coastal and marine areas worldwide are managed for conservation or protection [24]. Furthermore, a synthesis of human impact research on the ocean shows that no area is unaffected by human influence and over 40% are strongly affected by multiple drivers [2].

## 3. Materials and methods

### 3.1. Study area

GRNMS is an MPA approximately 20 miles (32 kilometers) off the coast of Georgia, USA in the Atlantic Ocean. It was designated as a national marine sanctuary in 1981 and covers 22 square miles (57 square kilometers), including one of the largest nearshore live-bottom reefs of the southeastern United States [10,25]. GRNMS is comprised of "scattered sandstone rock outcroppings that rise above the sandy substrate of the nearly flat continental shelf" [10], and includes soft corals, non-reef-building hard corals, and attached bivalves and sponges, as well as associated fishes, sea turtles, marine mammals, and pelagic birds. The location is at the intersection of temperate and tropical waters [10,25] at the South Atlantic Bight, and is influenced by the Gulf Stream, as well as tidal currents, river runoff, local winds, winter storms, hurricanes, and seasonal atmospheric changes [16]. GRNMS is an important habitat for several fish species that are commonly targeted by recreational anglers [10,26], and recreational fishers were found to be the dominant user group in the sanctuary [10,26,27]. However, fishing techniques are limited to rod and reel or handline fishing inside the sanctuary. Besides the main attraction of recreational fishing and fishing tournaments [10,26,27], diving is another ocean recreation activity possible in GRNMS for experienced divers. In addition to recreational use, the sanctuary is also frequently used for research. Since 2011, one-third of GRNMS has been a designated research area, where other uses are restricted [10,25].

## 3.2. Survey distribution

In the fall of 2020, the research team conducted an online survey focused on both users and non-users of GRNMS and surrounding coastal areas of Georgia [28]. The present study focuses specifically on users of GRNMS. The survey was distributed via Qualtrics to potential participants following the Dillman web surveying method [29]. In order to address the four traditional sources of survey errors, sampling, coverage, measurement and nonresponse [29], the survey distribution process was adjusted and done in a series of steps. After a first trial and feedback by different stakeholders, the first distribution of the survey was done with a smaller number of recipients to test the response rate. There is a risk the recipient of the email would not be interested in participating or forget to finish their survey. To minimize these risks, as suggested in the method [29], a first reminder email to the unfinished respondents was sent one week later, followed by a second two weeks later. The first distribution of the survey was sent on August 21, 2020. The emails that were sent to the contacts included a link to the survey (Qualtrics) and a description of the purpose of the project and the data collection.

## 3.3. Sample profile

The participants in this study were persons who held a saltwater permit fishing license in the state of Georgia in 2019. Contacts were obtained through and derived from the Georgia Department of Natural Resources angler license database. Potential respondents were contacted based upon their selection of the Saltwater Information Program permit registration in the state of Georgia. The sample of this study represents actual users of GRNMS, identified through a selective question in the survey instrument. Users of GRNMS are defined as the respondents that reported a visit to the sanctuary at least once in 2019 (N = 99). The socio-demographic data collected included the respondent's age, gender, race, ethnicity, education level, income level, and employment status.

## 3.4. Measures

The survey instrument described and used for this study builds upon previous surveys used by NOAA. The questionnaire was developed in response to a request by GRNMS marine resource managers (NOAA) with the intention to better understand resource users' knowledge, attitudes, and perceptions regarding reef health and management practices in GRNMS. The complete survey consisted of seven sections and 48 questions. The first section addressed the participants' opinions about protection and management of ocean and coastal resources, the second section asked about methods for communicating with potential resource users and the sources of information they use and trust. Section three aimed to identify people's opinion on the condition of resources and status of pressures in GRNMS. Sections four and five recorded visitors' recreation activities in the ocean, coastal areas, and GRNMS. The sixth section focused on ways participants value ocean and coastal resources and the marine environment. The goal was to learn about the ways respondents value products and services that are derived from ocean and coastal resources and the things they would do to help ensure their sustainability for the future. The last section addressed the participants' socio-demographic information.

For this study, specific survey questions about people's support of resource protection in GRNMS and coastal Georgia were used, in addition to a question about the extent to which people are willing to act to support sustainable use of resources. The demographic questions included information about place of residency, concern levels, and how people value ocean resources or services. The survey items used a five-point Likert-type scale; for example, when asking about support for protection, possible responses included: (1) No Support at All, (2) Somewhat Against, (3) Neutral, (4) Somewhat Support, (5) Strongly Support. To assess the potential impact of people's place of residency, two groups of respondents were identified based on the reported ZIP codes. Respondents were categorized into coastal or non-coastal ZIP codes. Coastal ZIP codes were defined as those bordering the coastline or within a 100-kilometer (approximately a one-hour drive) radius.

## 3.5. Data analyses

The database with the survey responses was exported from Qualtrics as an SPSS file and then analyzed using IBM SPSS (Statistical Package for the Social Sciences, IBM Corporation) version 28. To answer research question R1: "Do visitors support the protection of the ocean and resources in coastal Georgia and in GRNMS?" the descriptive statistics (frequencies, means, median, standard deviation, and standard error of mean) of the reported rankings of support of protection in GRNMS and coastal Georgia, as well as the reported extent to which respondents would be willing to take action to ensure that ocean and coastal resources are used sustainably were computed. The statistical tests aimed to measure the samples underlying attitudes toward protection in the study area. To analyze broader attitudes and the variance in the collected data, index variables were computed, grouping related response items together. Cronbach's alpha for all indices was computed and examined to measure the reliability of the indices or how well the variance in the indices correlated [29]. In order to answer research question R2: "What impacts the resource-users support of protection?" several statistical tests were used. A multi-model inference approach was used to understand how the combination of different predictor variables related to the reported scores of the support of protection. Using multi-model inference and Akaike information criterion (AIC) [30] scores allowed for comparison of different models against the information contained in the data [31]. Instead of relying on one model and testing it for significance, this analysis shows what combinations of predictors (based on the models that were used) give the most information relative to each other and the actual response variable of interest [31]. Different linear regression models were run and the AICc score was computed. The AICc score is a small-sample version of AIC, that includes a second-order bias correction [31]. The models were then compared and ranked based on the lowest AICc score of the model. The selection of the "best" model using AIC is done by choosing the lowest AIC scores and models with fewer parameters [31]. In order to answer research question R3: "To what extent are the participants willing to take actions to ensure a sustainable use of ocean resources?" different linear regression models (Table 2) were run once more, comparing the AICc score with different predictor variables. The models are described below.

Environmental attitudes and perceptions are predictors for environmentally friendly behavior and are linked to socio-demographics [20]. The multi-model inference analysis first aims to understand what factors the best predictors for the supportive attitudes across strata are, and second, what influenced their reported willingness to show certain environmentally friendly behavior. The model creation was driven by expert opinion, existing findings in the literature, and the research questions. The models were created with a maximum of four independent variables, to avoid overfitting the model. The variable selection for the models can be related to a more biocentric or anthropocentric value orientation. It is assumed that the way a person relates themselves to the environment can influence their concerns, their environmental values, their support of protection and their intention for sustainable behavior.

A metanalysis by Lotze et al. [4] focusing on public marine perceptions illustrated that pollution, fishing, climate change, biodiversity loss, and habitat degradation were perceived in most regions worldwide as the highest threats to marine ecosystems. In the same study, the authors found income and education were significant predictors for the public perception of marine protection among socio-demographic groups [4]. These variables were then used as a basis for some of the predictor variables in the models for the multi model inference analysis (models M1, M2, M6, M7, M11, M12). As a measure for the perception of threats the concern levels of related items were used either separately or combined into indices. For the model comparison five models were created, and the predictor variables were applied to three different dependent variables. The latter was represented by support of protection, willingness to pay/financial support, and consumption habits.

Furthermore, previous research has found a relationship between a positive environmental attitude and environmentally responsible behavior [3]. Environmental concern as an environmental attitude, as well as the willingness to act to ensure a sustainable use of marine resources, was expected to influence the respondent's support of protection and vice versa. Accordingly, these variables were included in combination with the place of residency in some of the models, such as model M5, M10, and M15, for the multi model inference analysis. The place of residency impacts how people feel and think about the environment [32–34] and the support of marine protection specifically [35].

 

In addition, socio-demographic variables influence how people perceive the environment [32] and were often found to be related to the perception or support of environmental protection [22]. Likewise, certain value orientations are associated with environmental concern and environmental behavior [14,19,21]. How well the socio-demographics or values predict the willingness to show sustainable behavior (M8, M9, M13, and M14) or support of protection (M3, M4) compared to other predictors was assessed in the analysis.

## 4. Results

The survey distribution resulted in 928 responses, of which around 99 (11%) of the respondents reported visiting GRNMS within the calendar year of 2019. More than half (58%) of the respondents were living in a coastal ZIP code, meaning within a 62 mile (100 km) radius of the coast (Table 1). Most respondents indicated their place of residency, based on their ZIP code, as the state of Georgia. Three quarters of respondents were over 50 years old at the time they took the survey.

Table 1. Demographic characteristics of survey respondents.

| Demographic | n | % |
|---|---|---|
| Age | | |
| 30 or younger | 2 | 5.1 |
| 31-50 | 8 | 20.5 |
| 51 or older | 29 | 74.4 |
| Gender | | |
| Male | 33 | 84.6 |
| Female | 6 | 15.4 |
| Race | | |
| White | 36 | 94.7 |
| Black/African American | 2 | 5.3 |
| Latino or Latina | 39 | 100.0 |
| Education | | |
| 9th–12th grade, no diploma | 1 | 2.5 |
| 12th grade high school graduate | 5 | 12.5 |
| Some college | 11 | 27.5 |
| Associate's degree | 5 | 12.5 |
| Bachelor's degree | 11 | 27.5 |
| Master's degree | 5 | 12.5 |
| Professional school degree | 1 | 2.5 |
| Doctor's degree | 1 | 2.5 |
| Income | | |
| Under $50,000 | 4 | 10.5 |
| $50,000–99,999 | 15 | 39.5 |
| Over $100,000 | 19 | 50.0 |
| Place of residency | | |
| Coastal ZIP Code | 22 | 57.9 |
| Non-coastal ZIP Code | 16 | 42.1 |
| Employment status | | |
| Full time | 27 | 71.1 |
| Part time | 1 | 2.6 |
| Retired | 9 | 23.7 |
| None of the above | 1 | 2.6 |

Only two respondents (5%) were 30 years of age or younger. On average, respondents were 58.05 years old. The median age was 62 years of age, meaning that half of the respondents were 62 years old or older. The standard deviation was 14 years. Male respondents (85%) outnumbered female respondents. Most people surveyed (95%) classified themselves as white, and the rest classified themselves as Black or African American; no other categories were reported regarding race. On average, respondents were employed full time, with a high annual household income. More than two-thirds of respondents (71%) reported a full-time employment status, and 23% reported being retired. Half of respondents selected one of the two highest income categories ($100,000–150,000 or more) for their annual household income before taxes in 2019. Most of the interviewees reported some college degree (27%) or a bachelor's degree (27%) as their highest completed level of education.

## 4.1. Support for protection

Overall, respondents reported a supportive attitude toward protection in the study area (Table 2). The mean for support of protection was 4.01 for users in coastal Georgia and ocean areas surrounding GRNMS and 4.22 for users inside of GRNMS. The results show that 76% of the respondents support protection in the entirety of the study area. In detail, approximately two-thirds of respondents indicated their support for protection associated with GRNMS surrounding areas in the "somewhat support" (33%) to "strongly support" (40%) range. Similarly, the results show that more than half of the surveyed users (52%) "strongly support" protection inside GRNMS, followed by another quarter (26%) being "somewhat" supportive of protection inside GRNMS. A little over 7% of the respondents expressed that they did not support protection in GRNMS or surrounding areas.

## 4.2. Willingness to act

The question about the extent to which respondents would be willing to undertake activities or action to ensure that the ocean and coastal resources are used sustainably and available for future generations to enjoy listed 10 different items. These included volunteering, contributing or supporting financially, including paying higher prices, taxes, fees or donating, or behavioral factors like adjusting consumption habits, such as recycling, using less energy or avoiding certain seafood products. The last item was "others" for respondents to fill out something they would do that was not listed. There were four responses to the open ended "other" item. Responses to that item included the mention of a political opinion instead of a potential action they would be willing to do. Additionally, the answer "voting for conservationists," "go fishing," and

**Table 2. Respondents' support for protection inside and outside of Gray's reef national marine sanctuary.**

| Area | Scale | Frequency (n) | Valid Percent (%) | M | SD | SE | N |
|------|-------|---------------|-------------------|---|----|----|---|
| Support for protection outside of GRNMS | 1- No Support at All | 3 | 4.3 | 4.01 | 1.056 | 0.126 | 70 |
| | 2- Somewhat Against | 2 | 2.9 | | | | |
| | 3- Neutral | 14 | 20 | | | | |
| | 4- Somewhat Support | 23 | 32.9 | | | | |
| | 5- Strongly Support | 28 | 40 | | | | |
| Support for protection inside GRNMS | 1- No Support at All | 1 | 1.4 | 4.22 | 0.989 | 0.116 | 73 |
| | 2- Somewhat Against | 4 | 5.5 | | | | |
| | 3- Neutral | 11 | 15.1 | | | | |
| | 4- Somewhat Support | 19 | 26 | | | | |
| | 5- Strongly Support | 38 | 52.1 | | | | |

*Note: M, SD, SE represent mean, standard deviation, standard error of mean and total responses, respectively.*

"see government practice more fiscal control" were given. These responses were not included in further analyses due to the small sample size.

The actions respondents were most willing to take were recycling (mean = 3.70), followed by using less energy (mean = 3.31) (Table 3). The lowest mean was reported for the action of donating to groups that represent diving interests. Furthermore, the frequency distribution along the scoring scale of the responses showed that volunteering and paying higher prices or taxes were the least prioritized without any rating for the highest score "would do the maximum" and with a higher frequency in the medium or lowest score. Most respondents were less willing to contribute in terms of paying more or donating. The scores indicating the willingness to pay higher taxes or to donate to groups representing diving interests show the lowest median (2.0), meaning that more than half of the responses were rated the lowest for these items. The highest median was reported for willingness to recycle, meaning that half or more of the respondents rated the highest two scoring levels for this item.

The reported responses regarding the extent to which the participants were willing to undertake action overall, regardless of the item, showed that most people rated "would do some" extent (32.6% of the ratings), followed by "would not do" (24.2%) (Fig 1). The least frequency of ratings considering all actions was made for "would do the maximum" (9%). Meaning, that in the sample the users of GRNMS most commonly "would do some" to ensure sustainable use of resources.

### 4.3. Multi model inference – Model selection

Responses for all variables used in the multi-model inference analyses were normally distributed. All indices had a satisfactory Cronbach's alpha between 0.638 and 0.963, showing that the items grouped together are closely related. After creating the models and examining the suitability for measuring underlying attitudes/ideologies using Cronbach's alpha, the regression analysis (including the AIC analysis) resulted in a successful model selection for further analyses in next steps. Comparing the AICc scores of all three model sets showed that M5, M10, and M15 were the best models with the lowest AICc score (Table 4, Table 5, Table 6).

The model comparison for set 1, including support of protection as the dependent variable, resulted in the selection of M5 as the best performing model based on the lowest AICc (129.04) value (Table 4). This model includes the index variable summarizing the overall concern of respondents, their overall willingness to perform sustainable behavior, and their place of residency. The combination of the included predictor variables was the best fit for explaining the support for protection in the sample. Other models do not need to be considered for interpretation based on the AICc results. Following Burnham et al. [31], competing models with delta AIC scores smaller than 2 units difference to the best fit model do not

**Table 3. Respondents' willingness to undertake action to ensure that ocean and coastal resources are used sustainably and available for future generations to enjoy.**

| Item/Action | M | MD | SD | SE | N |
|---|---|---|---|---|---|
| Volunteer time | 2.80 | 3.0 | 0.883 | 0.140 | 40 |
| Pay higher taxes | 2.18 | 2.0 | 1.035 | 0.164 | 40 |
| Pay higher prices for goods and services | 2.43 | 3.0 | 1.083 | 0.171 | 40 |
| Pay user fees | 2.60 | 3.0 | 1.081 | 0.171 | 40 |
| Donate to recreational fishing groups | 2.63 | 3.0 | 1.213 | 0.192 | 40 |
| Donate to diving groups | 1.90 | 2.0 | 1.057 | 0.167 | 40 |
| Recycle | 3.70 | 4.0 | 1.285 | 0.203 | 40 |
| Use less energy | 3.31 | 3.0 | 1.239 | 0.198 | 39 |
| Avoid/boycott certain seafood products | 2.75 | 3.0 | 1.463 | 0.231 | 40 |

*Note: M, MD, SD, SE represent mean, median, standard deviation, standard error of mean and total responses, respectively. Responses reported on a scale of 1 (No Support at All) to 5 (Strongly Support).*

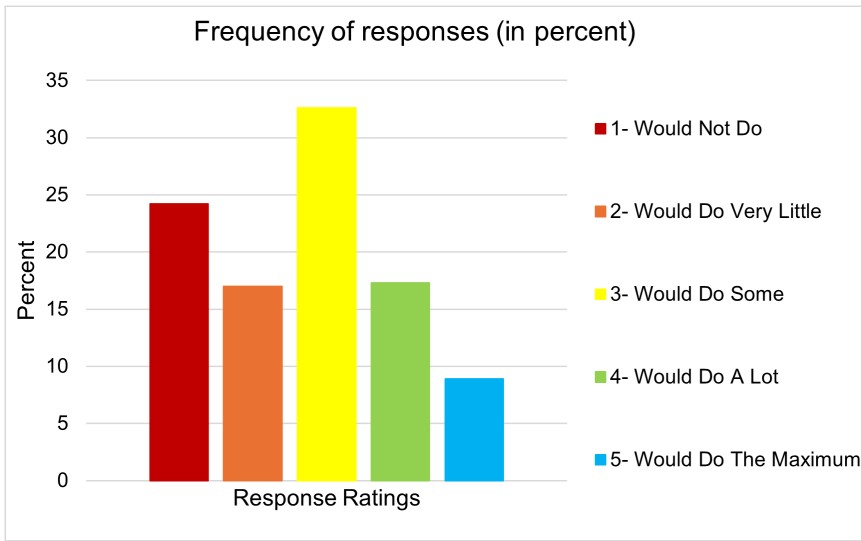

**Fig 1. Willingness to undertake action.**

**Table 4. Multi model inference analysis output table for model set 1 (support for protection).**

| Model | K | AIC$_c$ | Delta AIC$_c$ | AIC$_c$ Wt | LL |
|---|---|---|---|---|---|
| M5: Protection & concern, sustainable behavior, residency | 5 | 129.04 | 0 | 1 | −58.41 |
| M3: Protection & value | 5 | 152.99 | 23.96 | 0 | −70.56 |
| M4: Protection & socio-demographics | 6 | 168.11 | 39.08 | 0 | −76.66 |
| M2: Protection & concern, income, education | 17 | 189.03 | 59.99 | 0 | −61.41 |
| M1: Protection & concerns | 6 | 248.41 | 119.38 | 0 | −117.51 |
| null | 2 | 288.31 | 159.27 | 0 | −142.07 |

Note: Model selection was based on AIC$_c$. Lowest AIC$_c$ scores indicate best model fit. K, AIC$_c$, Delta AIC$_c$, AIC$_c$ Wt, and LL represent the number of parameters, AIC corrected, difference in AIC score between the best model and the model being compared, AIC$_c$ weight (total amount of predictive power), and log-likelihood, respectively.

**Table 5. Multi model inference analysis output table for model set 2 (willingness to pay).**

| Model | K | AIC$_c$ | Delta AIC$_c$ | AIC$_c$ Wt | LL |
|---|---|---|---|---|---|
| M10: Willingness to pay & concern, protection, residency | 5 | 191.7 | 0 | 0.99 | −89.78 |
| M8: Willingness to pay & values | 5 | 202.99 | 11.29 | 0 | −95.56 |
| M6: Willingness to pay & concerns | 6 | 204.31 | 12.61 | 0 | −94.75 |
| M7: Willingness to pay & concern, income, education | 5 | 217.4 | 25.7 | 0 | −102.76 |
| M9: Willingness to pay & socio-demographics | 6 | 220.02 | 28.32 | 0 | −102.65 |
| null | 2 | 220.89 | 29.19 | 0 | −108.29 |

Note: Model selection was based on AIC$_c$. Lowest AIC$_c$ scores indicate best model fit. K, AIC$_c$, Delta AIC$_c$, AIC$_c$ Wt, and LL represent the number of parameters, AIC corrected, difference in AIC score between the best model and the model being compared, AIC$_c$ weight (total amount of predictive power), and log-likelihood, respectively.

**Table 6. Multi model inference analysis output table for model set 3 (consumption habits).**

| Model | K | AIC$_c$ | Delta AIC$_c$ | AIC$_c$ Wt | LL |
|---|---|---|---|---|---|
| M15: Consumption habits & concern, protection, residency | 5 | 129.04 | 0 | 1 | −58.41 |
| M11: Consumption habits & concern | 5 | 152.99 | 23.96 | 0 | −70.56 |
| M13: Consumption habits & values | 6 | 168.11 | 39.08 | 0 | −76.66 |
| M12: Consumption habits & concern, income, education | 17 | 189.03 | 59.99 | 0 | −61.41 |
| M14: Consumption habits & socio-demographics | 6 | 248.41 | 119.38 | 0 | −117.51 |
| null | 2 | 288.31 | 159.27 | 0 | −142.07 |

Note: Model selection was based on AIC$_c$. Lowest AIC$_c$ scores indicate best model fit. K, AIC$_c$, Delta AIC$_c$, AIC$_c$ Wt, and LL represent the number of parameters, AIC corrected, difference in AIC score between the best model and the model being compared, AIC$_c$ weight (total amount of predictive power), and log-likelihood, respectively.

show enough predictive strength, meaning that the multi-model inference analysis does not support for these models for further consideration.

To test which model best predicts the respondent's willingness to contribute financially, a similar set of models was used and the AICc of each model was calculated. The lowest AICc showed model M10 (191.7), including overall environmental concern, overall reported support of protection, and place of residency as predictor variables (Table 5). This means that this combination of variables did the best job of explaining the willingness to pay out of all tested models and no other model needs to be selected based on the AICc results.

Finally, the third set of models were compared using the AICc value. The analysis resulted in the model selection of model M15 as the best fit model for explaining the willingness to adjust consumption habits such as recycling, using less energy, or avoiding certain seafood products. The lowest AICc showed this model with a value of 129.04 (Table 6). The combination of overall environmental concern, overall support of protection, and place of residency were the best performing predictors in terms of explaining potential for adjusting certain sustainable behaviors.

## 4.4. Best fit model – Regression

After the model selection based on AICc scoring, the statistical regressions of the best fit models were analyzed further. The results of the multiple regressions of each model are shown in in Table 7.

These analyses show that model M5 was significant ($p < 0.001$, R2 adj = 0.490) overall. The other two models were not significant. Additionally, the predictor variables showed some significant relationship with the reported support of

**Table 7. Results for multiple regressions of best fit models M5, M10, and M15, and p-values for variables.**

| Model | Sig. | Adjusted R Square | Predictor variables (x) | | | |
|---|---|---|---|---|---|---|
| | | | Overall Concern | Overall Willingness to Take Action | Place of Residency | Overall Support Protection |
| M5: Protection & concern, sustainable behavior, residency** | <0.001 | 0.490 | <0.001 *+ | 0.218 | 0.066- | N.A. |
| M10: Willingness to pay & concern, protection, residency | 0.118 | 0.093 | 0.391 | N.A. | 0.427 | 0.029*+ |
| M15: Consumption habits & concern, protection, residency | 0.903 | −0.082 | 0.522 | N.A. | −0.486 | 0.822 |

* significant predictor variable (p < 0.05)

** significant regression model (p < 0.05)

+ positive relationship (positive standardized coefficients beta)

- negative relationship (negative standardized coefficients beta)

protection. Overall concern (p < 0.001, standardized coefficients Beta β = 0.734) showed a significant predictive strength and a positive relationship to support of protection (Fig 2), meaning that the higher the environmental concern was, the higher the support of protection was rated.

Furthermore, place of residency (coastal/non-coastal) (p = 0.066, standardized coefficients Beta β = -0.263) was a non-significant variable; however, there was some evidence that there could be a relationship to support for protection (p-value < 0.1) (Table 7 and Fig 3). People living closer to the coast showed slightly higher support for protection than respondents living in non-coastal ZIP codes.

In order to assess drivers of respondents' willingness to perform a certain sustainable behavior, multiple linear regressions were run with model M10 (p = 0.118, R2 adj = 0.093) and M15 (p = 0.903, R2 adj = -0.082). M10 includes one significant positive relationship between the willingness to support sustainable use of the resources financially and the reported overall support of protection in the whole study area (p = 0.029, standardized coefficients Beta β = 0.513) (Fig 4).

## 5. Discussion

Resource management in marine protected areas is confronted with the challenge of balancing use and conservation of resources. Insights about visitor perceptions of management decision-making and planning can positively impact the success of environmental protection. Because MPAs such as GRNMS lack entry gates or infrastructure and are accessed via diverse entry and exit points to visit ocean areas, where intercepting visitors is possible, visitor use monitoring is challenging [26,36]. In such areas, data related to visitation can be insufficient. Specifically, in GRNMS, data about user profiles was lacking. This paper sought to: 1) understand the level of support for marine protection among users of GRNMS and surrounding coastal Georgia, 2) understand what behaviors they would engage in to ensure sustainable use of ocean resources, and 3) examine what drives and influences support for protection and willingness for activism.

Understanding resource users' and the broader public's perception can lead to more successful management outcomes [21]. Public familiarity with GRNMS and support for its protection could be enhanced by creating and expanding visitor centers, installing signage at departure points, and outreach and educational programs [25].

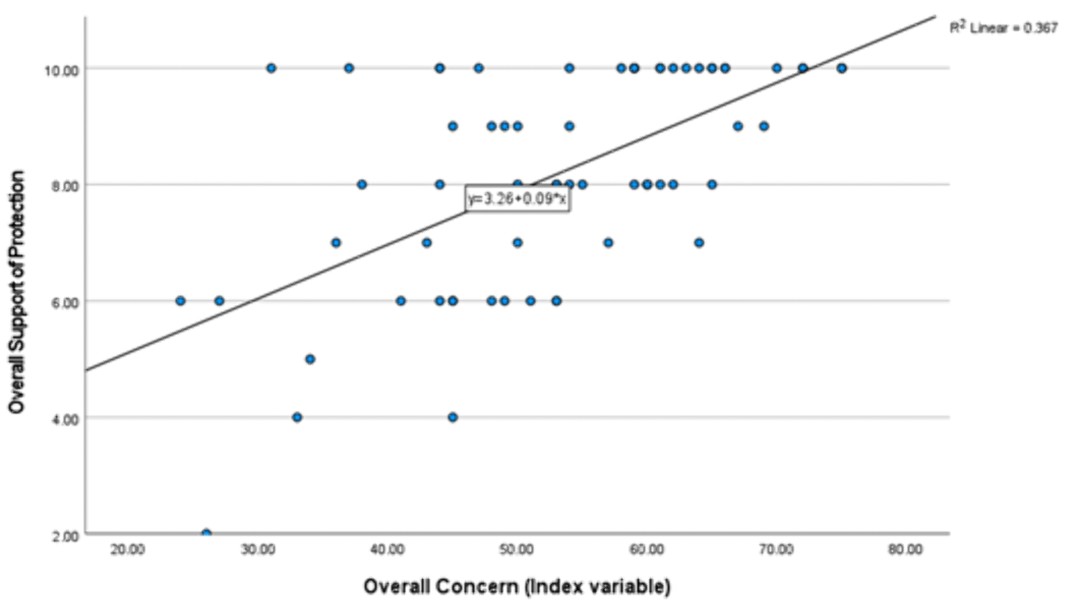

**Fig 2. Regression of support for protection and overall concern (M5).**

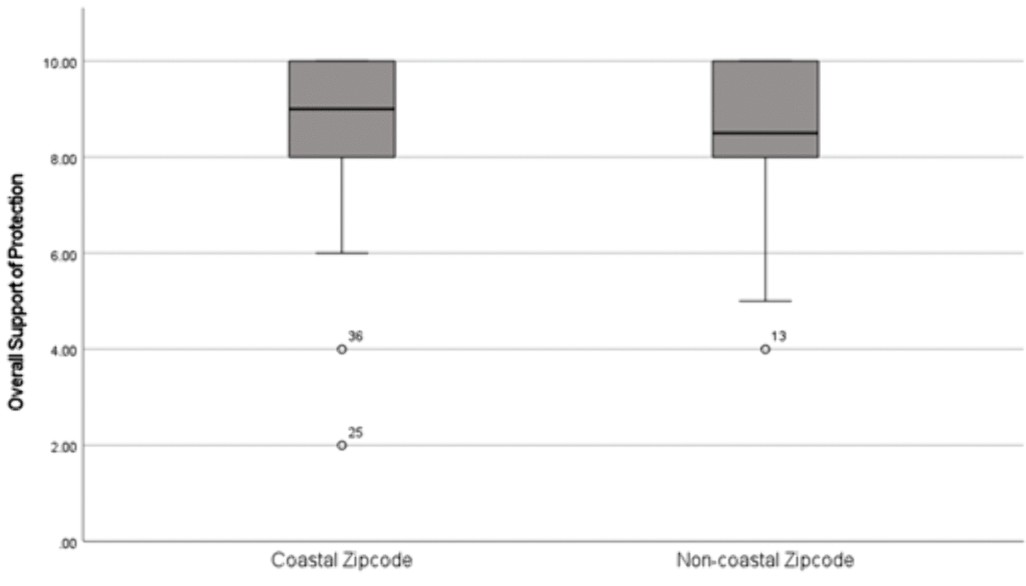

**Fig 3. Boxplot illustrating the relationship between place of residency and overall support for protection (M10).**

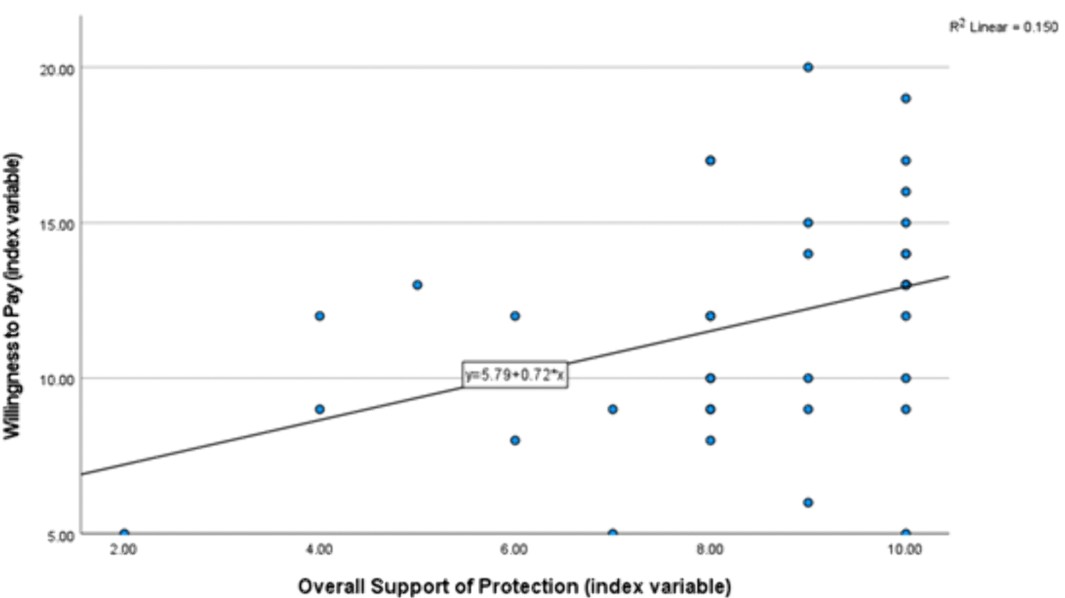

**Fig 4. Regression of overall support for protection and willingness to support sustainable use of resources financially (M10).**

Overall, respondents reported relatively moderate support for protection of GRNMS and surrounding coastal Georgia. The also results showed that willingness to take action was slightly lower than overall support for protection. The preferred action to ensure sustainable use of ocean resources was recycling, followed by using less energy, volunteering,

and avoiding certain seafood products. The least preferred group of items were associated with financial contribution and paying higher fees or taxes. These findings are generally inconsistent with existing literature regarding willingness to pay. However, willingness to take action has been linked to a person's attitudes and perception of consequences on an individual or societal level, as well as on the natural environment [37,38]. Previous research suggests that public support for protection can be improved by increasing the positive perception of social impacts and positive perception of governance related to protected area management [39]. To examine which predictor variables influenced the support for protection and the willingness to engage in sustainable behavior in the present study, a multi-model inference analysis was conducted. The results showed that the models including place of residency, in combination with overall index variables, were the best performing models over those that included socio-demographic predictors or specific values associated with ocean services or goods, or specific and common environmental concern items separately. The index variables represented overall environmental concern levels, overall willingness for action, and overall support for protection instead of ratings per separate item. Overall, the results of the analysis highlight the complexity of perception and intention for action/behavior. That means that support for protection is influenced by broader attitudes, such as environmental concerns, in combination with determinants like place of residency. For managers, it is beneficial to understand perception and attitude patterns of resource users and include these insights in their management decisions [4,7,11,13,39]. The results show the need for managers to consider and address the broader public and local communities, who are affected by potential restrictions resulting from a given area's protection status [13].

### 5.1. Limitations

A potential limitation of this study in regard to the influence of place of residency is the distance from the coast used to define coastal and non-coastal ZIP codes. Potts et al. [40] used a smaller distance of 20 km from the coast to define coastal areas in an assessment of European marine environmental perceptions. A 100 km radius was chosen in the present study because only the ZIP code, not exact address or location, was collected in the survey instrument, meaning that depending on the size of the ZIP code area, respondents may have been further away or closer to the coast. Due to varying size and shape of the areas represented as ZIP codes, a larger scale was used. Future studies should consider whether a smaller spatial scale affects the results reported here and if the differences would be more marked. Additionally, the remote survey method has some weaknesses [29]. We do not have specific information about non-respondents and people that were not reached by this survey [29]. Furthermore, the sample selection includes actual users of GRNMS, which might have impacted their reported perception, knowledge, and attitudes regarding marine environmental issues, resources, and protection. Additionally, survey contacts were selected based on their saltwater fishing license. Even though the identified primary recreational activity in GRNMS is recreational fishing [10,26,27], the sample might not cover the perceptions of other people who do not hold a fishing license. Additionally, the sample size was small (N = 99), which limits the ability to generalize these results to a larger group of users. However, despite this small sample size, this study was able to start filling a knowledge gap about the socio-demographic characteristics of potential visitors and resource users.

## 6. Conclusions

Environmental concern and attitudes can lead to environmentally friendly behavior [3]. Therefore, identifying concern levels among users can provide insights that can aid GRNMS resource managers in developing and implementing policies with higher chances of success [25,39,41,42]. At the same time, these data can illustrate the need for action and adaptive management to increase potential users' awareness and provide information and knowledge in areas where it might be lacking [19].

A primary objective of this study was to identify and describe the support for protection of ocean resources in and around GRNMS as well as the willingness to engage in specific sustainable behavior to ensure the conservation of ocean and coastal resources. Overall, the data showed relatively moderate support of protection of GRNMS and surrounding coastal Georgia, although willingness to take action was slightly lower. These data highlight the complexity of human behavior, a factor that sanctuary resource managers will need to account for when seeking to understand community support for ocean protection. Visitor use and tourism can provide tremendous incentive to support environmental protection by providing economic benefits and engaging different sectors of the community [5]. Increased support for environmental protection and engagement in environmentally responsible and sustainable behavior can in turn aid in resource conservation and reduce negative environmental impacts [7]. This study therefore suggests that including the human dimension and people in finding solutions for environmental problems is inevitable and essential, and thus understanding people's attitudes and perceptions can be greatly beneficial.

## Supporting information

**S1 File. GRNMS database clean.**
(CSV)

## Author contributions

**Conceptualization:** Robert Clyde Burns, Jasmine Cardozo Moreira, Danielle Schwarzmann.

**Data curation:** Marieke Lemmen.

**Formal analysis:** Robert Clyde Burns, Marieke Lemmen, Ross Andrew, Mary Allen.

**Funding acquisition:** Robert Clyde Burns, Ross Andrew, Danielle Schwarzmann.

**Investigation:** Robert Clyde Burns, Marieke Lemmen, Danielle Schwarzmann.

**Methodology:** Robert Clyde Burns, Ross Andrew, Danielle Schwarzmann.

**Project administration:** Robert Clyde Burns, Danielle Schwarzmann.

**Resources:** Danielle Schwarzmann.

**Supervision:** Robert Clyde Burns, Jasmine Cardozo Moreira.

**Writing – original draft:** Marieke Lemmen.

**Writing – review & editing:** Robert Clyde Burns, Ross Andrew, Jasmine Cardozo Moreira, Mary Allen.

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
