## [Decision Letter · Decision Letter 0]

19 Jun 2025

Dear Dr. Burns,

Thank you for submitting your manuscript to PLOS ONE. After careful consideration, we feel that it has merit but does not fully meet PLOS ONE’s publication criteria as it currently stands. Therefore, we invite you to submit a revised version of the manuscript that addresses the points raised during the review process.

**The reviewers have provided insightful comments and suggestions that we believe will enhance the quality and clarity of your work. We kindly encourage you to address these points in your revised submission, as they are designed to strengthen both the scientific rigor and overall impact of the manuscript.**

We look forward to receiving your revised manuscript.

Kind regards,

A S Sochipem Zimik

Academic Editor

PLOS ONE

**Journal Requirements:**

1. When submitting your revision, we need you to address these additional requirements. Please ensure that your manuscript meets PLOS ONE's style requirements, including those for file naming. The PLOS ONE style templates can be found at https://journals.plos.org/plosone/s/file?id=wjVg/PLOSOne_formatting_sample_main_body.pdf and https://journals.plos.org/plosone/s/file?id=ba62/PLOSOne_formatting_sample_title_authors_affiliations.pdf 2. In the ethics statement in the methods, you have specified that verbal consent was obtained. Please provide additional details regarding how this consent was documented and witnessed, and state whether this was approved by the IRB. 3. We note that the grant information you provided in the ‘Funding Information’ and ‘Financial Disclosure’ sections do not match.  When you resubmit, please ensure that you provide the correct grant numbers for the awards you received for your study in the ‘Funding Information’ section. 4. In the online submission form, you indicated that “The data underlying the results presented in the study are available from the author”.  All PLOS journals now require all data underlying the findings described in their manuscript to be freely available to other researchers, either a. In a public repository, b. Within the manuscript itself, or c. Uploaded as supplementary information.This policy applies to all data except where public deposition would breach compliance with the protocol approved by your research ethics board. If your data cannot be made publicly available for ethical or legal reasons (e.g., public availability would compromise patient privacy), please explain your reasons on resubmission and your exemption request will be escalated for approval. 5. Please amend either the abstract on the online submission form (via Edit Submission) or the abstract in the manuscript so that they are identical. 6. Please include your full ethics statement in the ‘Methods’ section of your manuscript file. In your statement, please include the full name of the IRB or ethics committee who approved or waived your study, as well as whether or not you obtained informed written or verbal consent. If consent was waived for your study, please include this information in your statement as well. 7. We note that Figure 1 in your submission contain map images which may be copyrighted. All PLOS content is published under the Creative Commons Attribution License (CC BY 4.0), which means that the manuscript, images, and Supporting Information files will be freely available online, and any third party is permitted to access, download, copy, distribute, and use these materials in any way, even commercially, with proper attribution. For these reasons, we cannot publish previously copyrighted maps or satellite images created using proprietary data, such as Google software (Google Maps, Street View, and Earth). For more information, see our copyright guidelines: http://journals.plos.org/plosone/s/licenses-and-copyright. We require you to either present written permission from the copyright holder to publish these figures specifically under the CC BY 4.0 license, or remove the figures from your submission: a. You may seek permission from the original copyright holder of Figure 1 to publish the content specifically under the CC BY 4.0 license.   We recommend that you contact the original copyright holder with the Content Permission Form (http://journals.plos.org/plosone/s/file?id=7c09/content-permission-form.pdf) and the following text:“I request permission for the open-access journal PLOS ONE to publish XXX under the Creative Commons Attribution License (CCAL) CC BY 4.0 (http://creativecommons.org/licenses/by/4.0/). Please be aware that this license allows unrestricted use and distribution, even commercially, by third parties. Please reply and provide explicit written permission to publish XXX under a CC BY license and complete the attached form.” Please upload the completed Content Permission Form or other proof of granted permissions as an "Other" file with your submission. In the figure caption of the copyrighted figure, please include the following text: “Reprinted from [ref] under a CC BY license, with permission from [name of publisher], original copyright [original copyright year].” b. If you are unable to obtain permission from the original copyright holder to publish these figures under the CC BY 4.0 license or if the copyright holder’s requirements are incompatible with the CC BY 4.0 license, please either i) remove the figure or ii) supply a replacement figure that complies with the CC BY 4.0 license. Please check copyright information on all replacement figures and update the figure caption with source information. If applicable, please specify in the figure caption text when a figure is similar but not identical to the original image and is therefore for illustrative purposes only.The following resources for replacing copyrighted map figures may be helpful: USGS National Map Viewer (public domain): http://viewer.nationalmap.gov/viewer/The Gateway to Astronaut Photography of Earth (public domain): http://eol.jsc.nasa.gov/sseop/clickmap/Maps at the CIA (public domain): https://www.cia.gov/library/publications/the-world-factbook/index.html and https://www.cia.gov/library/publications/cia-maps-publications/index.htmlNASA Earth Observatory (public domain): http://earthobservatory.nasa.gov/Landsat: http://landsat.visibleearth.nasa.gov/USGS EROS (Earth Resources Observatory and Science (EROS) Center) (public domain): http://eros.usgs.gov/#Natural Earth (public domain): http://www.naturalearthdata.com/ 8. Please include your tables as part of your main manuscript and remove the individual files. Please note that supplementary tables (should remain/ be uploaded) as separate "supporting information" files. 9. We note that this data set consists of interview transcripts. Can you please confirm that all participants gave consent for interview transcript to be published? If they DID provide consent for these transcripts to be published, please also confirm that the transcripts do not contain any potentially identifying information (or let us know if the participants consented to having their personal details published and made publicly available). We consider the following details to be identifying information:- Names, nicknames, and initials- Age more specific than round numbers- GPS coordinates, physical addresses, IP addresses, email addresses- Information in small sample sizes (e.g. 40 students from X class in X year at X university)- Specific dates (e.g. visit dates, interview dates)- ID numbers Or, if the participants DID NOT provide consent for these transcripts to be published:- Provide a de-identified version of the data or excerpts of interview responses- Provide information regarding how these transcripts can be accessed by researchers who meet the criteria for access to confidential data, including:a) the grounds for restrictionb) the name of the ethics committee, Institutional Review Board, or third-party organization that is imposing sharing restrictions on the datac) a non-author, institutional point of contact that is able to field data access queries, in the interest of maintaining long-term data accessibility.d) Any relevant data set names, URLs, DOIs, etc. that an independent researcher would need in order to request your minimal data set. For further information on sharing data that contains sensitive participant information, please see: https://journals.plos.org/plosone/s/data-availability#loc-human-research-participant-data-and-other-sensitive-data If there are ethical, legal, or third-party restrictions upon your dataset, you must provide all of the following details (https://journals.plos.org/plosone/s/data-availability#loc-acceptable-data-access-restrictions):a) A complete description of the datasetb) The nature of the restrictions upon the data (ethical, legal, or owned by a third party) and the reasoning behind themc) The full name of the body imposing the restrictions upon your dataset (ethics committee, institution, data access committee, etc)d) If the data are owned by a third party, confirmation of whether the authors received any special privileges in accessing the data that other researchers would not havee) Direct, non-author contact information (preferably email) for the body imposing the restrictions upon the data, to which data access requests can be sent 10. Please include captions for your Supporting Information files at the end of your manuscript, and update any in-text citations to match accordingly. Please see our Supporting Information guidelines for more information: http://journals.plos.org/plosone/s/supporting-information.

Reviewers' comments:

Reviewer's Responses to Questions

**Comments to the Author**

1. Is the manuscript technically sound, and do the data support the conclusions?

Reviewer #1: Yes

Reviewer #2: Partly

2. Has the statistical analysis been performed appropriately and rigorously?

Reviewer #1: Yes

Reviewer #2: Yes

3. Have the authors made all data underlying the findings in their manuscript fully available?

Reviewer #1: Yes

Reviewer #2: Yes

4. Is the manuscript presented in an intelligible fashion and written in standard English?

Reviewer #1: Yes

Reviewer #2: Yes

**Reviewer #1: ** General comments: The manuscript addresses a timely and relevant topic concerning public support for ocean protection in a marine protected area (MPA), specifically Gray's Reef National Marine Sanctuary (GRNMS). The study is well-structured and aligns with conservation priorities, several areas require improvement. But I have itemized some important comments below that the authors should attend to; in areas where the request was already included but I didn’t catch it, please point those out.

Specific comments:

1. The title could be more precise by specifying the region studied (e.g. “....at Gray’s Reef National Marine Sanctuary, Georgia”), especially since the study emphasizes spatial variation in perceptions.

2. The abstract is overly descriptive and lacks specificity in methodology and key statistical results. For example, the phrase “relatively high support” should be backed with exact figures or mean scores. It would benefit from including the statistical tools used (e.g. multi-model inference, regression analysis) and one or two concrete numerical results I think.

3. The introduction overuses general claims without proper citations or logical flow in parts. For instance, the second sentence, “Thus, the connections people have to the ocean, individually and as a society, are numerous,” sounds generic to me. It would be stronger if the authors cited empirical studies illustrating these connections. Similarly, while the paradox of tourism is discussed, the introduction does not clearly articulate the unique research gap this study fills, especially in comparison with other studies of U.S. marine sanctuaries.

4. The research questions (lines 254, 264, and 277) are buried in the methods. These should be explicitly stated at the end of the introduction to guide the reader. Their current placement dilutes their impact.

5. The paragraph beginning “Understanding environmental values…..” (lines 134) repeats ideas already introduced in 2.1 without contributing new synthesis. This could be removed or integrated more succinctly.

6. The manuscript states that “users of GRNMS are defined as the respondents that reported a visit within 2019” (line 202). However, the initial pool was selected based on fishing license databases. Could this lead to sampling bias, as it excludes non-fishing users like divers or non-consumptive recreationists?

7. The Dillman web method is mentioned, but the authors do not explain how their adjustments were tailored to the study’s context. For example, was any pilot testing done to validate the cultural or regional appropriateness of the questions? Were reminders standardized across all rounds?

8. The sample profile suggests major demographic skewness. The sample is overwhelmingly male (85%) and white (95%). This limitation should be addressed more explicitly in both the discussion and limitations sections, as it strongly impacts the generalizability of the findings.

9. The survey measures section refers to seven sections and 48 questions, yet only those relevant to “support of protection” and “willingness to act” are discussed in the analysis. What happened to the rest? A flowchart or schematic summarizing which items contributed to which indices would improve transparency.

10. It is problematic that only one regression model (M5) was statistically significant, and that adjusted R² values for the other two models were low (M10 = 0.093; M15 = -0.082). I think these weak models should be discussed with more caution, especially given that conclusions are drawn from them. Relying heavily on non-significant findings can mislead the interpretation of the drivers of willingness to act.

11. Table 1 is misleading in presentation. It states that “N = 39” for age but later says total respondents were 99. This discrepancy should be clarified. I don’t get it there, were only 39 respondents included in the socio-demographic analysis?

12. Table 3 uses a 5-point scale but combines responses in a way that may obscure interpretation. For example, reporting means and medians for actions like recycling and donating do not capture the actual behavioral intention variability. Boxplots or histograms would be more informative.

13. The discussion overstates the results in several areas. It claims “relatively high support of protection,” but 24% of respondents expressed no or low support, which is not negligible. Similarly, it says respondents “would do some,” yet the overall willingness to act was modest and skewed away from high-effort behaviors (e.g.. donations, volunteering).

14. The statement that ‘place of residency showed some evidence of a relationship to support” (line 421, ) is based on a p-value of 0.066, which is not statistically significant. So the language should be more cautious here and avoid overstating borderline results.

15. The idea of stewardship is discussed without clearly defining it or showing how it was operationalized in this study. If stewardship is a central theme, the authors should show how survey responses reflect stewardship principles (e.g., personal responsibility, conservation ethics).

16. The conclusion restates the entire paper without providing strong synthesis or actionable insights. It could be shortened and focused more on practical implications for marine sanctuary managers and future survey development.

17. The limitations section is quite strong but appears too late in the manuscript. It might be more impactful if merged with the discussion or presented earlier. The issue of relying solely on anglers for sampling should be emphasized more clearly, as it systematically excludes non-consumptive users. Maybe put it before the conclusion.

**Reviewer #2: ** This is an interesting paper, however, based solely on literature research and statistics. The biggest weakness of the paper is the lack of a research methodology section. I suggest supplementing the paper with such a section and adding a diagram/schema showing the research process.

**Do you want your identity to be public for this peer review?** For information about this choice, including consent withdrawal, please see our Privacy Policy

Reviewer #1: No

Reviewer #2: No

---

## [Author Response · Author response to Decision Letter 1]

9 Jul 2025

Response to Reviewers

Dear Reviewers: Thank you for your thought and suggestions regarding this manuscript. I have revised the paper accordingly and have outlined a response below.

Specific comments:

1. The title could be more precise by specifying the region studied (e.g. “....at Gray’s Reef National Marine Sanctuary, Georgia”), especially since the study emphasizes spatial variation in perceptions. Thank you, the title was changed to include Gray’s Reef National Marine Sanctuary.

2. The abstract is overly descriptive and lacks specificity in methodology and key statistical results. For example, the phrase “relatively high support” should be backed with exact figures or mean scores. It would benefit from including the statistical tools used (e.g. multi-model inference, regression analysis) and one or two concrete numerical results I think. Thank you. Abstract was revamped to discuss the study more specifically.

3. The introduction overuses general claims without proper citations or logical flow in parts. For instance, the second sentence, “Thus, the connections people have to the ocean, individually and as a society, are numerous,” sounds generic to me. It would be stronger if the authors cited empirical studies illustrating these connections. Similarly, while the paradox of tourism is discussed, the introduction does not clearly articulate the unique research gap this study fills, especially in comparison with other studies of U.S. marine sanctuaries. Thank you. Parts of the introduction were re-written to reduce the reliance on author notes.

4. The research questions (lines 254, 264, and 277) are buried in the methods. These should be explicitly stated at the end of the introduction to guide the reader. Their current placement dilutes their impact. Thank you. The research questions were added as recommended.

5. The paragraph beginning “Understanding environmental values…..” (lines 134) repeats ideas already introduced in 2.1 without contributing new synthesis. This could be removed or integrated more succinctly. Section 2.3 focuses specifically on marine environmental values and goes deeper than the introduction in section 2.1. Thank you. This recommendation was followed by a revision of the text. Section 2.1 is designed to preface sections 2.2 and 2.3.

6. The manuscript states that “users of GRNMS are defined as the respondents that reported a visit within 2019” (line 202). However, the initial pool was selected based on fishing license databases. Could this lead to sampling bias, as it excludes non-fishing users like divers or non-consumptive recreationists? Thank you. While non-consumptive use may occur in this extremely low-use setting, what little recreation occurs tends to be fishing. This was addressed within the manuscript in several places.

7. The Dillman web method is mentioned, but the authors do not explain how their adjustments were tailored to the study’s context. For example, was any pilot testing done to validate the cultural or regional appropriateness of the questions? Were reminders standardized across all rounds? Thank you. Yes, this was addressed in the lines above and below 202.

8. The sample profile suggests major demographic skewness. The sample is overwhelmingly male (85%) and white (95%). This limitation should be addressed more explicitly in both the discussion and limitations sections, as it strongly impacts the generalizability of the findings. Thank you. Recreation use in this off-shore ocean sanctuary is primarily males who are fishing. The study would not be generalizable against most other sanctuary settings.

9. The survey measures section refers to seven sections and 48 questions, yet only those relevant to “support of protection” and “willingness to act” are discussed in the analysis. What happened to the rest? A flowchart or schematic summarizing which items contributed to which indices would improve transparency. Thank you, discussion regarding the seven sections was deleted.

10. It is problematic that only one regression model (M5) was statistically significant, and that adjusted R² values for the other two models were low (M10 = 0.093; M15 = -0.082). I think these weak models should be discussed with more caution, especially given that conclusions are drawn from them. Relying heavily on non-significant findings can mislead the interpretation of the drivers of willingness to act. Thank you. Text was changed to address this important issue.

11. Table 1 is misleading in presentation. It states that “N = 39” for age but later says total respondents were 99. This discrepancy should be clarified. I don’t get it there, were only 39 respondents included in the socio-demographic analysis? Thank you. Unfortunately the data provided included only socio-demographic information for 40 respondents.

12. Table 3 uses a 5-point scale but combines responses in a way that may obscure interpretation. For example, reporting means and medians for actions like recycling and donating do not capture the actual behavioral intention variability. Boxplots or histograms would be more informative. Thank you, the means and medians are reported in a way that is standard for similar published manuscripts.

13. The discussion overstates the results in several areas. It claims “relatively high support of protection,” but 24% of respondents expressed no or low support, which is not negligible. Similarly, it says respondents “would do some,” yet the overall willingness to act was modest and skewed away from high-effort behaviors (e.g.. donations, volunteering). Thank you, this important point was outlined on Line 349, where we show that 7% did not support,

14. The statement that ‘place of residency showed some evidence of a relationship to support” (line 421, ) is based on a p-value of 0.066, which is not statistically significant. So the language should be more cautious here and avoid overstating borderline results. Thank you, text was revised accordingly.

15. The idea of stewardship is discussed without clearly defining it or showing how it was operationalized in this study. If stewardship is a central theme, the authors should show how survey responses reflect stewardship principles (e.g., personal responsibility, conservation ethics). Thank you, the discussion of stewardship was deleted.

16. The conclusion restates the entire paper without providing strong synthesis or actionable insights. It could be shortened and focused more on practical implications for marine sanctuary managers and future survey development. Thank you, the conclusion was shortened and focused more on implications for marine sanctuary resource management.

17. The limitations section is quite strong but appears too late in the manuscript. It might be more impactful if merged with the discussion or presented earlier. The issue of relying solely on anglers for sampling should be emphasized more clearly, as it systematically excludes non-consumptive users. Maybe put it before the conclusion. Thank you, the limitations were moved and now emphasize the high proportion of anglers.

---

## [Decision Letter · Decision Letter 1]

25 Aug 2025

Dear Dr. Burns,

Thank you for submitting your manuscript to PLOS ONE. After careful consideration, we feel that it has merit but does not fully meet PLOS ONE’s publication criteria as it currently stands. Therefore, we invite you to submit a revised version of the manuscript that addresses the points raised during the review process.

**Dear authors**
**Thank you for answering the comments from the reviewers. Please refer to the comments of Reviewer 1 and submit a modified manuscript. **
**Additionally, there are some aspects that I would like to point out:**

Please check for typos, extra spaces, and lack of full stops (e.g., L39) in the manuscriptI agree with the reviewer, there is a need for an introductory part in the abstract.Chapter 2 is really a literature review, or more like a background.Several sub-chapters are not numbered (e.g., L208) after 3.2 or Model Creation after 3.4Line 221: from here is a description of the sample profile. Why here?Line 235: why only 39 respondentsIn Table 1, what is the Latino or Latino and then No? Makes little sense to me. Also, in Table 1, why are there 40 valid answers for Education when it was said that 39 was the sample size?Finally, the conclusions should be integrated throughout the discussion. Conclusions are meant to give an overview of the results obtained and lay the foundations for future research.

We look forward to receiving your revised manuscript.

Kind regards,

Miguel Inácio

Academic Editor

PLOS ONE

Journal Requirements:

Reviewers' comments:

Reviewer's Responses to Questions

**Comments to the Author**

Reviewer #1: (No Response)

Reviewer #2: All comments have been addressed

2. Is the manuscript technically sound, and do the data support the conclusions?

Reviewer #1: Yes

Reviewer #2: Yes

3. Has the statistical analysis been performed appropriately and rigorously?

Reviewer #1: Yes

Reviewer #2: Yes

4. Have the authors made all data underlying the findings in their manuscript fully available?

Reviewer #1: Yes

Reviewer #2: Yes

5. Is the manuscript presented in an intelligible fashion and written in standard English?

Reviewer #1: Yes

Reviewer #2: Yes

Reviewer #1: I believe the abstract section still needs some focused refinement. At the moment, it jumps directly into stating the objective without first offering a clear introductory context or problem statement. Starting with a concise and compelling sentence that frames the broader issue would help orient the reader before moving into the purpose of the work.

Additionally, I noticed a few typos, such as "us" instead of "use," as well as some long or inconsistent sentences that could be rephrased for clarity. The strength of a good abstract lies in its ability to deliver a powerful, succinct summary that immediately captures the interest of the target audience. In its current form, I think the abstract doesn’t quite achieve that yet, but with a few revisions particularly in structure and language, it can be significantly improved.

Reviewer #2: Thank you for completing this paper. At the moment, the value of the paper is significantly higher. This affects both the quality of the data presented in the paper and the overall evaluation of the journal.

**Do you want your identity to be public for this peer review?** For information about this choice, including consent withdrawal, please see our Privacy Policy

Reviewer #1: No

Reviewer #2: No

---

## [Author Response · Author response to Decision Letter 2]

28 Sep 2025

Please note that the responses to reviewers/editor are included in the uploaded files, dated Sep 28, 2025

---

## [Editor Report · Decision Letter 2]

8 Oct 2025

Dear Dr. Burns,

Thank you for submitting your manuscript to PLOS ONE. After careful consideration, we feel that it has merit but does not fully meet PLOS ONE’s publication criteria as it currently stands. Therefore, we invite you to submit a revised version of the manuscript that addresses the points raised during the review process.

**ACADEMIC EDITOR:**
**Dear authors, thank you for addressing all the queries. **
**However, before accepting the manuscript, I would like to clarify again the issue of the 39, 40 and 38 respondents. The authors responded that this was explained in the text. However, I do not find this explanation. Could you please explain why the analysis was conducted for only 39 respondents, and in the tables, the numbers of respondents for some categories are listed as 38 and 40? Please clarify this, since this has been an issue in previous versions as well. This is intended to clarify the matter for the manuscript's readers. Please address this. **

**Additionally, consider keeping limitations as a separate sub-chapter within the discussion. And if the authors do not want to include a conclusion, then at least a sub-chapter as concluding remarks.**

We look forward to receiving your revised manuscript.

Kind regards,

Miguel Inácio

Academic Editor

PLOS ONE
---

## [Author Response · Author response to Decision Letter 3]

16 Oct 2025

Please see responses to reviewer dated 10.16.2025

---

## [Editor Report · Decision Letter 3]

21 Oct 2025

Visitors’ Support of Ocean Protection in a Low Use Marine Protected Area: Gray's Reef National Marine Sanctuary

PONE-D-25-19622R3

Dear Dr. Burns,

We’re pleased to inform you that your manuscript has been judged scientifically suitable for publication and will be formally accepted for publication once it meets all outstanding technical requirements.

Kind regards,

Miguel Inácio

Academic Editor

PLOS ONE

Additional Editor Comments (optional):

Dear authors,

Thank you for addressing all queries.

Perhaps just modify in the Proofs phase the chapter 5.2 Conclusions to 6. Conclusions.
---

## [Editor Report · Acceptance letter]

PONE-D-25-19622R3

PLOS ONE

Dear Dr. Burns,

I'm pleased to inform you that your manuscript has been deemed suitable for publication in PLOS ONE. Congratulations! Your manuscript is now being handed over to our production team.

Kind regards,

on behalf of

Dr. Miguel Inácio

Academic Editor

PLOS ONE